# CHARACTERIZE AND TRANSFER ATTENTION IN GRAPH NEURAL NETWORKS

## ABSTRACT

Does attention matter and, if so, when and how? Our study on both inductive and transductive learning suggests that datasets have a strong influence on the effects of attention in graph neural networks. Independent of learning setting, task and attention variant, attention mostly degenerate to simple averaging for all three citation networks, whereas they behave strikingly different in the protein-protein interaction networks and molecular graphs: nodes attend to different neighbors per head and get more focused in deeper layers. Consequently, attention distributions become telltale features of the datasets themselves. We further explore the possibility of transferring attention for graph sparsification and show that, when applicable, attention-based sparsification retains enough information to obtain good performance while reducing computational and storage costs. Finally, we point out several possible directions for further study and transfer of attention.

## 1 INTRODUCTION

The modeling of graphs has become an active research topic in deep learning (Bronstein et al., 2017). Dozens of neural network models have been developed for exploiting the structural information of graphs (Scarselli et al., 2009; Bruna et al., 2014; Henaff et al., 2015; Duvenaud et al., 2015; Niepert et al., 2016; Defferrard et al., 2016), now collectively referred to as graph neural networks (GNNs).

Built upon the success of attention in NLP (Vaswani et al., 2017), Veličković et al. (2018) proposed the graph attention networks (GATs) to integrate multi-head *self-attention* into node feature update for adaptive weighting, with several extensions (Thekumparampil et al., 2018; Zhang et al., 2018; Monti et al., 2018; Svoboda et al., 2019; Trivedi et al., 2019). While the use of attention in GNNs is an attractive direction, several works also report that attention contributes little to the performance of GNNs (Zhang et al., 2018; Shchur et al., 2019). Considering the high computational cost of attention, the question is then that does attention help and, if so, when and how?

In this paper, we take a first step towards the question. We first identify the key questions for understanding attention and propose an analytical paradigm. With extensive experiments, our findings suggest that, although attention is motivated by inductive learning, its functionality depends highly on the characteristics of the datasets. The attention distributions across heads and layers are near uniform for all citation networks (*Cora*, *Citeseer* and *Pubmed*) while they get more concentrated over layers on the protein-protein interaction networks (*PPI*) and molecular graphs, with significant diversity among heads. That the attention distribution is a telltale sign of the nature of graph class is further verified with a meta graph classification experiment. With attention features as inputs, citation networks are indistinguishable whereas *PPI* and molecule graphs are.

Inspired by these findings, we hypothesize that attention carry semantic meanings when they are non-uniform and can be helpful for transfer learning. This has been the case in the NLP community (Radford et al., 2019), and is motivating many research efforts on understanding multi-head attention (Jain & Wallace, 2019; Clark et al., 2019; Voita et al., 2019). We attempt the idea of attention based sparsification – sparsifying a graph by retaining edges where attention are higher, with the intuition being that the resulting graph preserves enough information. We find that not only such attention-based sparsification is transferable (meaning, it can work on unseen graphs), it also affords us to train a cheaper model without using attention to fit the downstream task. Finally, we discuss several possible fruitful directions for further exploration, including theory, interpretability, and unsupervised learning.

## 2 RELATED WORK

**Visualize and understand attention** Several works attempted to visualize the learned attention by coloring edges or nodes based on the attention magnitudes (Veličković et al., 2018; Qiu et al., 2018). Thekumparampil et al. (2018) studied the averaged attention values between nodes with different or the same class labels. Shanthamallu et al. (2018) studied the attention GAT learned on two citation networks *Cora* and *Citeseer* with interquartile range metric and showed that they are near uniform. Knyazev et al. (2019) investigated the effectiveness of attentional pooling and find that attention is only effective when it is close to optimal. Our work differs in that we propose a general paradigm for analyzing graph attention, including how to characterize the overall attention statistics and how to measure the layer-wise and head-wise differences of the learned attention.

**Transfer attention** In the computer vision community, transferring the attention maps using a teacher-student network to improve the downstream tasks is a well-studied technique (Zagoruyko & Komodakis, 2017; Li et al., 2019). Our approach for transferring attention is different from these works in that we use the trained graph attention network as a graph sparsifier instead of a teaching signal. Yang et al. (2018) proposed to transfer the relational structure within the data, which is represented as a set of attention weights, to boost the performance of other tasks. Our transferring strategy is different from their work because we reduce the sparsity of the affinity matrix by removing the entries with smaller attention weights. Thus, we can train a cheap graph sparsifier to accelerate the training and testing speed.

## 3 BACKGROUND

### 3.1 GRAPH NEURAL NETWORKS

Let $G$ be an undirected graph with node set $\mathcal{V}$ and edge set $\mathcal{E}$, where each node $i \in \mathcal{V}$ has a feature $h_i^0 \in \mathbb{R}^{n_0}$. In a wide class of GNNs (Kipf & Welling, 2017; Hamilton et al., 2017; Veličković et al., 2018), the basic feature update function for node $i \in \mathcal{V}$ at the $l + 1$-th GNN layer takes the form of

$$
h_i^{l+1} = \sigma \left( \sum_{j \in \mathcal{N}(i)} \alpha_{i,j}^{l+1} \mathbf{W}^{l+1} h_j^l \right),
$$

where $\sigma$ is an activation function, $\mathcal{N}(i)$ is a set containing $i$ and its neighbors, $\alpha_{i,j}^{l+1} \in \mathbb{R}$ is the attention weight of edge $(j, i)$ in updating the feature of node $i$, $\mathbf{W}^{l+1} \in \mathbb{R}^{n_{l+1} \times n_l}$ is the projection matrix, and $h_i^l, h_i^{l+1}$ are corresponding node features after the $l$-th and the $l + 1$-th layer. With a sparse implementation, it has a time complexity of $O(|\mathcal{V}|n_{l+1}n_l + |\mathcal{E}|n_{l+1})$.

**Graph Convolutional Network (GCN)** (Kipf & Welling, 2017) and the mean variant of **Graph-SAGE** (Hamilton et al., 2017) use static attention $\frac{1}{\sqrt{|\mathcal{N}(i)|}} \frac{1}{\sqrt{|\mathcal{N}(j)|}}$ and $\frac{1}{|\mathcal{N}(i)|}$, which we separately refer to as *GCN* and *uniform* attention.

**GAT** (Veličković et al., 2018) uses a parameterized subnetwork to output the attention weights $\alpha_{i,j}$s. Rather than using a single attention head as in Eqn. equation 3.1, GAT aggregates the outputs of multiple heads:

$$
\alpha_{i,j}^{l+1,k} = \frac{\exp\left(\text{score}\left(h_i^l, h_j^l\right)\right)}{\sum_{j' \in \mathcal{N}(i)} \exp\left(\text{score}\left(h_i^l, h_{j'}^l\right)\right)}, \qquad h_i^{l+1,k} = \sigma \left( \sum_{j \in \mathcal{N}(i)} \alpha_{i,j}^{l+1,k} \mathbf{W}^{l+1,k} h_j^l \right),
$$

$$
h_i^{l+1} = \sigma \left( \text{Aggregate}^{l+1} \left( h_i^{l+1,1}, \cdots, h_i^{l+1,K^{l+1}} \right) \right),
$$

where $k$ is the index of the attention head and $K^{l+1}$ is the number of attention heads in the $l + 1$-th layer. Aggregate$^{l+1}$ aggregates all head results in the $l + 1$-th layer and we follow the approach of Veličković et al. (2018) to use concatenation for intermediate layers and average for the final layer.

**Attention variants** As in Luong et al. (2015), there are multiple ways to calculate the attention scores. In this paper, we focus on the following three types of attention, namely *concat*, *dot product*,

Table 1: Dataset task and learning setting

|  |  | Cora | Citeseer | Pubmed | PPI | CEP | HIV |
|---|---|:---:|:---:|:---:|:---:|:---:|:---:|
| Task | Node Classification | ✓ | ✓ | ✓ | ✓ |  |  |
|  | Graph Prediction |  |  |  |  | ✓ | ✓ |
| Setting | Transductive Learning | ✓ | ✓ | ✓ |  |  |  |
|  | Inductive Learning |  |  |  | ✓ | ✓ | ✓ |

and *general*:

$$\text{LReLU}(\mathbf{a}^T[\mathbf{W}h_i||\mathbf{W}h_j]) \text{ (concat)}, \quad (\mathbf{W}h_i)^T\mathbf{W}h_j \text{ (dot product)}, \quad (\mathbf{W}h_i)^T\mathbf{B}\mathbf{W}h_j \text{ (general)}$$

GATs uses the *concat* attention. Zhang et al. (2018) and Ryu et al. (2018) separately explores *dot product* and *general* attention in GNNs. Also, $\text{LReLU}(\cdot)$ means the leaky ReLU activation.

**Graph-level prediction** Based on the node representations generated by GNNs, we can also compute a graph representation (Li et al., 2018) for graph-level prediction problems like graph classification and regression:

$$h_G = \sum_{v \in \mathcal{V}} \text{Sigmoid}\left(g\left(h_v^L\right)\right) \text{ReLU}\left(f\left(h_v^L\right)\right),$$

where $L$ is the number of GNN layers, $h_v^L$ is the representation of node $v$ output by the last layer of GNN, $g(\cdot)$ calculates the impact of node $v$ on the graph representation, and $f(\cdot)$ is a linear projection.

## 3.2 TASKS AND DATASETS

We consider the tasks of node classification and graph-level prediction. For modeling, we treat all graphs as undirected with untyped nodes and edges. Self loops are added to preserve information from previous node features. As in Veličković et al. (2018), we consider four datasets – citation networks *Cora*, *Citeseer* (Sen et al., 2008), *Pubmed* (Namata et al., 2012) and *PPI* (Zitnik & Leskovec, 2017). Additionally, we include two more datasets of molecular graphs for graph-level prediction.

The Harvard Clean Energy Project (CEP) (Hachmann et al., 2011) estimates the photovoltaic efficiency of organic molecules. We use a subset of it pre-processed by Duvenaud et al. (2015); Ryu et al. (2018). The HIV dataset was initially introduced by the Drug Therapeutics Program (DTP) AIDS Antiviral Screen HIV for testing the ability of compounds to inhibit HIV replication. It was later included in the MoleculeNet benchmark (Wu et al., 2018) as a binary classification task. For both datasets, the node features are extracted based on DeepChem (Ramsundar et al., 2019) and RDKit (Landrum), which includes atom type, degree, and many other chemical properties.

Since GAT was partially motivated to work on unseen data, we consider two learning settings: transductive learning and inductive learning. In the transductive learning setting, the model can access the features of all nodes in the graph. However, only a fraction of the nodes are labeled in the training phase and the model is asked to predict the missing labels. In the inductive learning setting, we have two mutually exclusive sets of nodes separately for training and testing. The model is trained only on the features and labels of the nodes in the training set and is asked to predict the labels of the nodes in the testing set. A summary of the tasks and learning settings can be found in Table 1. We leave more detailed information like dataset statistics, training/testing split and features in Appendix A.

## 4 METHODOLOGY

The introduction of multi-head attention into multi-layer GNNs poses many interesting questions, we investigate five in this paper. **Q1**: In the GAT model, all nodes have different attention distributions on their incoming edges. How should we characterize the overall statistics of these learned attention distributions? **Q2**: How do attention distributions differ across different heads and layers? **Q3**: How does the choice of dataset, attention variant, and learning setting affect the learned attention? **Q4**:

Table 2: Discrepancy between static attention and learned attention by the first head in the first layer

|                   | Cora   | Citeseer | Pubmed | PPI    | CEP    | HIV    |
|-------------------|--------|----------|--------|--------|--------|--------|
| uniform vs learned | 0.0083 | 0.0020   | 0.0059 | 0.5442 | 0.1754 | 0.2376 |
| GCN vs learned     | 0.1118 | 0.0796   | 0.1999 | 0.5791 | 0.1759 | 0.2258 |

Is the statistics of the learned attention related to the intrinsic properties of the graph? **Q5**: How to transfer attention for further usage?

To answer **Q1**, we propose multiple metrics for characterizing attention distributions. For **Q2**, we examine the metrics at different layers and compare the change of them over layers. To answer **Q3**, we run experiments to see how varying the dataset, attention variant and the learning setting impacts the learned attention. Previous works (Kipf & Welling, 2017; Hamilton et al., 2017; Veličković et al., 2018) only perform transductive learning on the citation networks and inductive learning on *PPI*. To fill in the gap, we perform transductive learning on *PPI* and inductive learning on the citation networks (see data processing strategy in Appendix B.1). We show that learning tasks are largely irrelevant. To answer **Q4**, we propose a new task called *Meta Graph Classification* which asks the model to distinguish the type of the graphs by the characteristics of the attention distributions. For **Q5**, we transfer attention for graph sparsification and examine whether we can preserve enough task-related information with a significant number of edges removed.

# 5 CHARACTERIZING ATTENTION

## 5.1 ANALYZING ATTENTION METRICS

**Experiment settings**    Our attention study is completely based on the GAT architecture except that we try different types of attention mentioned in Section 3.1. We follow the experiment settings of the original authors whenever possible and perform a hyperparameter search otherwise. To make a fair comparison between attention variants, we use the same hidden size for each layer output across attention variants. The detailed settings can be found in Appendix B.2. Unless explicitly mentioned, we perform 100 random runs for *Cora*, *Citeseer* and *Pubmed* and 10 random runs for *PPI*, *CEP* and *HIV*. The test performance is mostly consistent across attention variants and is comparable to the original reported numbers, which we report in Appendix B.4.

**Learned attention v.s. static attention**    Attention-free GNNs employ static weights in updating node features. The first question is then how does learned attention differ from static weights? If the learned attention are almost the same as the static ones, then there is no point in performing costly attention computation and we do not need to proceed with the analysis. For any static attention (*GCN* or *uniform*), we quantify the discrepancy between it and learned attention for node $i$ with $\frac{1}{2} \sum_{j \in \mathcal{N}(i)} |\alpha_{i,j}^{\text{learned}} - \alpha_{i,j}^{\text{static}}|$. With a range of $[0, 1]$, the larger the value, the larger the discrepancy is. We average this value over all nodes in graphs. Table 2 shows the discrepancy between the static and the learned attention for the first head with the '*concat*' variant. Surprisingly, the discrepancy against the uniform attention is very small for the citation networks, suggesting that each node attends similarly to different neighbors.

**Head-wise and layer-wise differences**    Figure 2 visualizes the attention of a node over its incoming edges in *Cora* and *PPI*, based on three heads in the last layer. We can find that different heads behave distinctively in the *PPI* case and they are all uniform in the *Cora* case. We summarize the change of attention over heads and layers with several metrics. To quantify the variance between two head-wise distributions, we compute the averaged $L_1$ norm of the difference between the mean distribution over all heads and the learned distribution for each head.

$$\alpha_{i,j}^{\text{mean}} = \frac{1}{K} \sum_{k=1}^{K} \alpha_{i,j}^k, j \in \mathcal{N}(i), \quad \text{Head-wise Variance} = \frac{1}{2K} \frac{1}{|\mathcal{V}|} \sum_{k=1}^{K} \sum_{i \in \mathcal{V}} \sum_{j \in \mathcal{N}(i)} |\alpha_{i,j}^k - \alpha_{i,j}^{\text{mean}}|.$$

To probe the concentration of attention, we compute the maximum pairwise difference of them within one-hop neighborhoods $\max_{j_1, j_2 \in \mathcal{N}(i)} |\alpha_{i,j_1} - \alpha_{i,j_2}|$. To verify whether attention are on self loops when they are concentrated and GNNs degenerate to MLPs, we monitor the self-loop attention values.

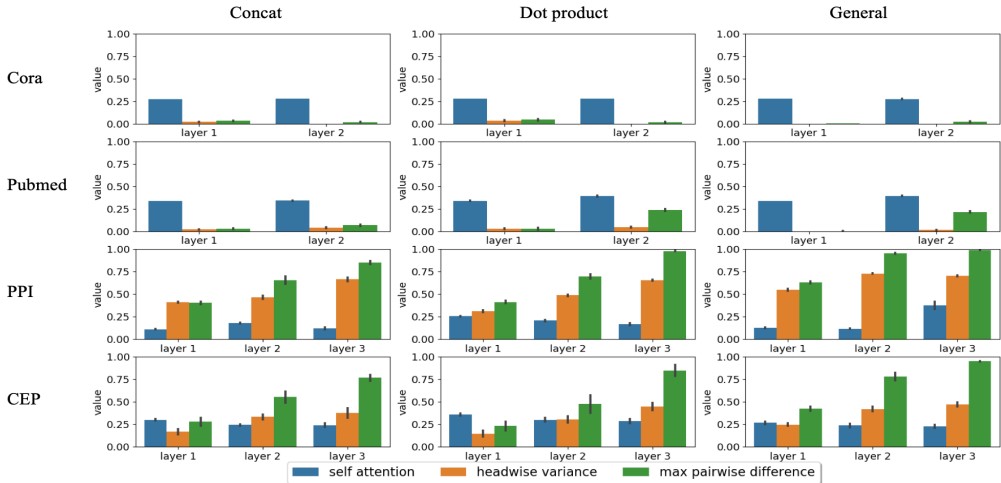

Figure 1: Comparison of the learned attention across layers, datasets and attention variants

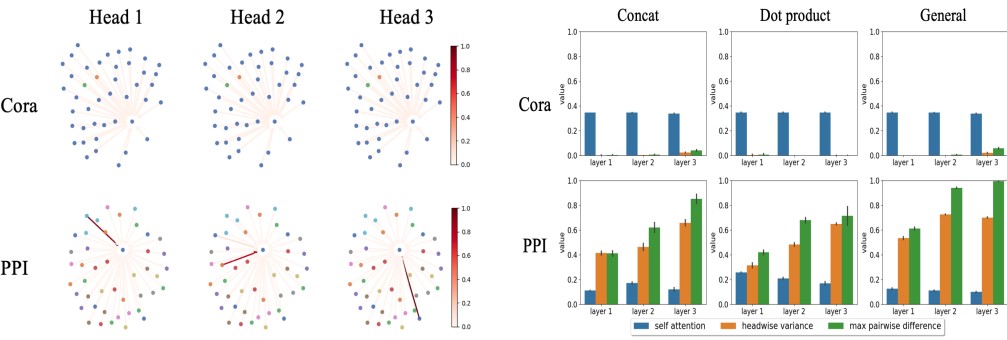

Figure 2: Visualization of the learned attention of one node. Nodes are colored by labels and edges are colored by attention magnitude.

Figure 3: Attention learned by training on Cora and PPI with inductive and transductive settings respectively.

We average the metrics over all nodes and heads for a layer in each run, and compute the mean and standard deviation of the averaged metrics in all runs.

**Varying settings and attention variants** We leverage the defined metrics to compare the learned attention by different attention variants. Figure 1 visualizes the head-wise and layer-wise metrics for *Cora*, *Pubmed*, *PPI* and *CEP*. Independent of the attention variant, the learned attention change little across layers for *Cora* and *Pubmed* while the increasing max pairwise difference indicates that they get more concentrated with deeper layers for *PPI* and *CEP*. Besides, the attention does not get increasingly more concentrated on self loops while getting sharper over layers. We also experiment with different training settings on these graphs. Figure 3 shows the learned attention when training inductively on *Cora* while training transductively on *PPI*. We observe the similar phenomenon of the learned attention, which eliminates the effect of training settings.

## 5.2 META GRAPH CLASSIFICATION

Previous experiments suggest that the attention learned are highly graph-dependent and their characteristics can be predicted with proper knowledge of graph semantics. To verify it, we attempt to infer the graph types based on the attention learned. Specifically, we perform graph classification with attention-based features.

Table 3: Graph Classification Accuracy

|  | Concat | General | Dot product |
|---|---|---|---|
| All Layers | $94.1 \pm 0.5\%$ | $95.6 \pm 0.7\%$ | $95.5 \pm 0.4\%$ |
| First Layer | $81.3 \pm 1.1\%$ | $88.4 \pm 0.8\%$ | $91.3 \pm 0.6\%$ |
| Second Layer | $83.5 \pm 0.6\%$ | $89.7 \pm 0.6\%$ | $85.3 \pm 0.5\%$ |

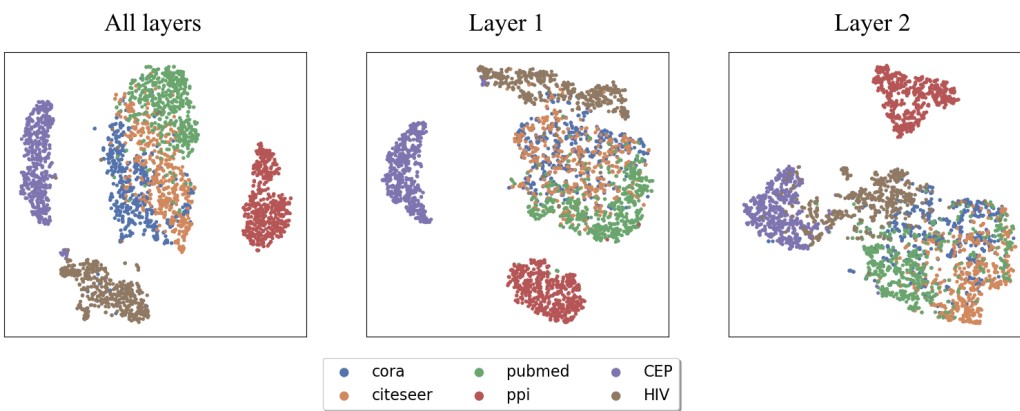

Figure 4: t-SNE visualization of '*concat*' attention based features. From left to right, the features are from all layers, the first, and the second layer, respectively. See Appendix B.5 for more results.

**Synthetic dataset**  We construct a synthetic dataset for graph classification with two steps. First, we collect 480 samples/subgraphs of 20 to 30 nodes from each dataset. For HIV and CEP, we choose 480 graphs and construct a balanced subset. For the rest four datasets, we sample 480 graphs for each using random walk as in the case of inductive learning on citation networks. Second, we separately train a 2-layer GAT for each collected dataset and compute the attention metrics for each layer. The mean and standard deviation of the metrics are then used as the graph features. We leave more details in Appendix B.5.

We train a logistic regression classifier for graph classification, where 20% of the graphs are used for training and the rest are used for testing. We have separately experimented with attention metrics from all layers, the first layer, and the second layer. The classification performance is reported in Table 3, with all experiments repeated for 10 times.

In the experiments, we find that more than 80% of the incorrect classifications happen within citation subgraphs. This is also verified by our t-SNE (van der Maaten & Hinton, 2008; Pedregosa et al., 2011) visualization of the attention metrics in Figure 4. We can see that the attention metrics of citation networks tend to be indistinguishable while the attention metrics of other datasets are better separated and clustered.

## 6  ATTENTION-BASED GRAPH SPARSIFICATION

### 6.1  PPI SPARSIFICATION FOR GAT PREDICTION

We consider two intuitive heuristics for attention based sparsification: 1) *local top-k* sparsification selects up to $k$ incoming edges for each node with highest attention values; 2) *global threshold* sparsification selects edges whose attention value exceeds a pre-specified threshold over the entire graph. Across all GNN layers, we perform attention-based sparsification for each attention head to get a subset of edges. We then take the union of the edge subsets to get the edge set for the sparsified graph. All self loops are selected to preserve the information of original node features.

Inspired by recent research on sampling based training of GNNs (Hamilton et al., 2017; Huang et al., 2018), we consider two baseline random sparsification for comparison: 1) *uniform neighbor* sparsification uniformly samples at most $k$ incoming edges for each node without replacement and we compare it against *local top-k* sparsification; 2) *uniform graph-wise* sparsification uniformly

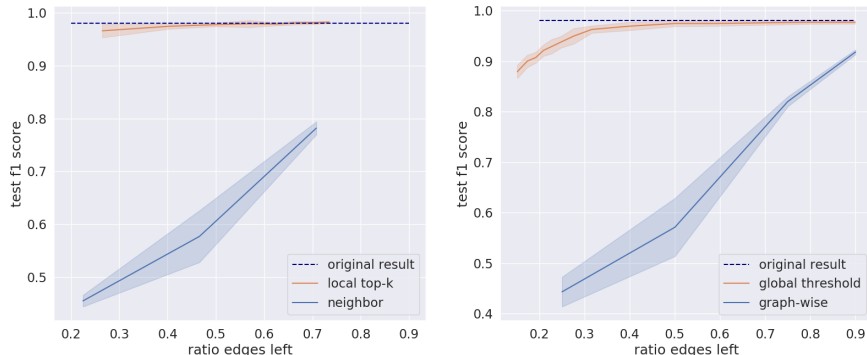

Figure 5: Comparison of attention based sparsification against baseline random sparsification. The result of training on unsparsified graphs is included for reference. For *local top-k* sparsification we consider $1 \leq k \leq 8$. For *global threshold* sparsification, we consider threshold in $\{0.9, 0.8, 0.7, 0.6, 0.5, 0.4, 0.3, 0.2, 0.1, 0.05, 0.03, 0.01, 0.005\}$.

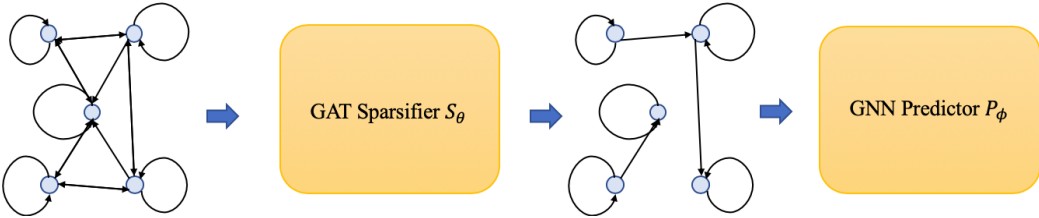

Figure 6: Sparsify graphs with one GAT and predict on sparsified graphs with another GNN

samples a proportion of edges over the entire graph(s) without replacement and we compare it against *threshold* sparsification.

We first experiment attention-based graph sparsification with *PPI*, which tends to have sharp attention-based on our previous study and is relatively dense (56944 nodes and 1644208 edges with self loops and bi-directional edges added). Despite similar sharp attention, molecules are not good candidates for the experiment as they are already sparse and on average each atom only has less than three neighbors. As illustrated in figure 6, after training a GAT on PPI, we perform attention-based sparsification and re-train a GAT on the sparsified training and validation graphs.

Figure 5 compares the results of attention-based sparsification with *concat* variant against random sparsification. 1) With a similar degree of sparsification, the attention-based sparsification consistently performs better than the baseline in terms of test metric and its variance across runs; 2) We can reach a test accuracy comparable to the original result with only $40\% \sim 50\%$ edges left in the training and validation graphs.

## 6.2    SPARSIFICATION WITH GENTLE ATTENTION

What if the attention are neither uniform nor sharp? Our previous study shows that the attention in *Pubmed* are not completely uniform in the last GAT layer with *dot product* and *general* attention variants so we use it as a testbed for the study. In cases where attention are not very sharp, *threshold* sparsification is not very useful and we consider only *top-k* sparsification.

Table 4 compares *top-k* sparsification with *dot product* attention against *uniform neighbor* sparsification. With similar proportion of edges left in the training and validation graphs, the *top-k* sparsification consistently outperforms *uniform neighbor* sparsification and achieves a performance comparable to that of training on the raw graphs.

Table 4: Attention-based sparsification against random sparsification on *Pubmed*

| Sparsification | % edges left | Test score |
|---|---|---|
| None | 100% | $78.20 \pm 0.70\%$ |
| *local top-k (k=1)* | $57.37 \pm 0.78\%$ | $77.57 \pm 0.12\%$ |
| *local top-k (k=2)* | $71.44 \pm 0.74\%$ | $\mathbf{78.33 \pm 0.29}\%$ |
| *uniform neighbor* | 59.13% | $74.40 \pm 0.36\%$ |
| *uniform neighbor* | 72.72% | $75.47 \pm 1.03\%$ |

Table 5: *local top-1* sparsificatiton with light GATs of different sizes followed by GraphSAGE prediction on *PPI*. For reference, GraphSAGE can reach a test score of $0.9802$ on unsparsified graphs and a test score of $0.5431$ on graphs sparsified by baseline methods with about $30\%$ edges left.

| # hidden | GAT test score | # GAT parameters | % edges left | GraphSAGE test score |
|---|---|---|---|---|
| 4 | 0.5208 | 27574 | 29.55% | 0.9700 |
| 16 | 0.7718 | 107686 | 30.06% | 0.9776 |
| 64 | 0.9704 | 520294 | 30.95% | **0.9802** |

## 6.3 LIGHT GAT SPARSIFICATION WITH GRAPHSAGE PREDICTION

For practical usage of speeding up the computation, we do not want to train two large GATs from scratch. We propose to first quickly train a light GAT with a lot fewer parameters for graph sparsification and then train a large attention-free GNN for prediction on the sparsified graphs.

We explore this idea by varying the hidden sizes of attention heads in GAT and use a GraphSAGE for prediction on *PPI*. The GraphSAGE has 3 layers and each layer has a hidden size of $512$. We consider the mean aggregator for it with skip connection added. The results are summarized in Table 5. Our experiments show that a GraphSAGE model can achieve a good performance with training on sparsified graphs as long as we sparsify test graphs. Surprisingly, while a smaller hidden size in GAT does harm its own prediction performance, it has little effect on capturing important edges for GraphSAGE prediction even with *top-1* sparsification. We note that the number of parameters in GraphSage and the baseline GAT is 1.2M and 3.6M, respectively, i.e. a 3-fold reduction.

## 7 CONCLUSIONS AND DISCUSSIONS

In this work, we propose an analytical paradigm that can summarize the characteristics of multi-head attention learned over graphs and compare them against topology-based static attention. This allows a deeper understanding of attention beyond simply comparing the performance of models and motivates further use of attention like transfer learning. In addition to the attention-based sparsification we explored, we believe below are several interesting directions that are underexplored:

- **Theory.** Many efforts have performed to theoretically understand and explain GNNs, particularly their connection to kernel methods and Weisfeiler-Lehman tests (Morris et al., 2019; Xu et al., 2019; Maron et al., 2019), but few of them have considered attention.

- **Interpretability.** The use of attention can add interpretability. This is particularly valued in risk-sensitive scenarios like medicine. Several efforts have been made in the chemistry community (Ryu et al., 2018; Preuer et al., 2019; Xiong et al., 2019).

- **Unsupervised learning.** Unsupervised representation learning is an important approach when the training of a model is expensive and the labeled data is scarce. This is particularly the case for biological networks and molecular graphs due to the need of wet-lab experiments. The NLP community has witnessed the success of unsupervised learning with attention (Radford et al., 2019) and we might expect the same for GNNs. Weihua Hu (2019) demonstrates the effectiveness of training GNNs with unsupervised learning for chemistry and biology, but did not employ attention.

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

# A  DATASETS

## A.1  DATASET SUMMARY

Table 6 and 7 summarize the statistics about the raw graph datasets. When computing the number of edges and node degrees, we have not considered self loops. Also when we model directed graphs as undirected graphs, the number of edges will get doubled. For transductive learning, the number of edges is considered to be the same for training/validation/test as all edges may be involved in message passing.

Table 6: Statistics and properties for single graph datasets

| Properties | Datasets | | |
| --- | --- | --- | --- |
| | Cora | Citeseer | Pubmed |
| # node feats | 1433 | 3703 | 500 |
| # classes/labels | 7 | 6 | 3 |
| # graphs | 1 | 1 | 1 |
| # nodes | 2708 | 3327 | 19717 |
| # isolates | 0 | 48 | 0 |
| # edges | 5278 | 4552 | 44324 |
| # mean degree | 1.95 | 1.37 | 2.25 |
| # train nodes | 140 (5.2%) | 120 (3.6%) | 60 (0.3%) |
| # train edges | 5278 | 4552 | 44324 |
| # val nodes | 300 (11.1%) | 500 (15.0%) | 500 (2.5%) |
| # val edges | 5278 | 4552 | 44324 |
| # test nodes | 1000 (36.9%) | 1000 (30.1%) | 1000 (5.1%) |
| # test edges | 5278 | 4552 | 44324 |

Table 7: Statistics and properties for multi-graph datasets

| Properties | Datasets | | |
| --- | --- | --- | --- |
| | PPI | CEP | HIV |
| # node feats | 50 | 58 | 75 |
| # classes/label size | 121 (binary multi-label) | 1 (regression) | 1 (binary) |
| # graphs | 24 | 29978 | 41913 |
| # nodes | 56944 | 829135 | 1069968 |
| # isolates | 256 | 0 | 2351 |
| # edges | 793632 | 1000952 | 1151942 |
| # average degree | 13.9 | 1.2 | 1.1 |
| # train graphs | 20 | 17986 (60.0%) | 33530 (80.0%) |
| # train nodes | 44906 (78.9%) | 497311 (60.0%) | 846823 (79.1 %) |
| # train edges | 613184 (77.3%) | 600365 (60.0%) | 906666 (78.7 %) |
| # val graphs | 2 | 5995 (20.0%) | 4191 (10.0%) |
| # val nodes | 6514 (11.4%) | 165916 (20.0%) | 117216 (11.0%) |
| # val edges | 99460 (12.5%) | 200269 (20.0%) | 128751 (11.2%) |
| # test graphs | 2 | 5997 (20.0%) | 4192 (10.0%) |
| # test nodes | 5524 (9.7%) | 165908 (20.0%) | 105929 (9.9%) |
| # test edges | 80988 (10.2%) | 200318 (20.0%) | 116525 (10.1%) |

## A.2  ADDITIONAL DETAILS

**Graph Construction**   For citation networks, the nodes correspond to documents and the edges correspond to citations between pairs of documents. For *PPI*, the nodes represent proteins and the edges represent physical interactions between them. For molecules, the nodes correspond to atoms and the edges correspond to chemical bonds.

**Node featurization and labeling**   For citation networks, nodes have bag-of-words features and labels for the topic of documents. For *PPI*, the node features include positional gene sets, motif gene sets and immunological signatures and the node labels are gene ontology sets (Hamilton et al., 2017), collected from the Molecular Signatures Database (Subramanian et al., 2005). For the CEP dataset, the node features consist of one hot encodings of atom type, node degree, the total number of hydrogens attached to it, the number of implicit hydrogens attached to it, and its aromaticity indicator. For the HIV dataset, in addition to those features we also consider the formal charge of the atom, the number of radical electrons of the atom and the atom's hybridization.

**Dataset splits**   For Cora, Citeseer, Pubmed, and PPI, we consider a deterministic dataset split for training, validation and test. For CEP, we randomly split the dataset in each run, where approximately the proportion of graphs used for training, validation and test is separately $60\%, 20\%$ and $20\%$. The statistics of the CEP dataset in table  7 is obtained in one random run. For the HIV dataset, we use the scaffold split (Wu et al., 2018; Li et al., 2017), which structurally separates molecules into training, validation and test subsets and poses a greater challenge for generalization.

**Imbalanced dataset**   The HIV dataset is highly imbalanced, with only 1487 compounds are positive, constitute approximately $3.5\%$ of the dataset.

## B  EXPERIMENT SETTINGS

### B.1  VARYING LEARNING SETTING FOR CITATION NETWORKS AND PPI

**Transductive Learning on *PPI***   To perform transductive learning on *PPI*, we sample two mutually exclusive subsets of the nodes as the training set and validation set for each graph, leaving the rest as the test set. We experiment on two splitting settings. In the first setting, we sample about $5\%$ nodes for training and $18\%$ nodes for validation, similar to the splitting ratio of the transductive learning setting on *Cora*. In the second setting, we sample $79\%$ nodes for training and $11\%$ nodes for validation, similar to the case of inductive learning on *PPI*.

**Inductive Learning on Citation Networks**   To perform inductive learning on citation networks, we first sample 120 graphs of 100 nodes for each dataset. We use a random walk based sampling algorithm described in Algorithm 1, which by the study of (Leskovec & Faloutsos, 2006) performs best in preserving the properties of static graphs. Separately, $60\%, 20\%, 20\%$ of the graphs are used for training, validation and test.

### B.2  HYPERPARAMETERS FOR ATTENTION STUDY

For the attention study, we consider the hyperparameters below:

- Transductive learning on Cora and Citeseer: 2-layer GAT with 8 heads in the first layer and 1 head in the second layer, 8 hidden units for each head in the first layer, a dropout of 0.6, no residual connection, a learning rate of 0.005, L2 regularization with coefficient 0.0005, cross entropy loss

- Transductive learning on Pubmed: 2-layer GAT with 8 heads in both layers, 8 hidden units for each head in the first layer, a dropout of 0.6, no residual connection, a learning rate of 0.01, L2 regularization with coefficient 0.001, cross entropy loss

- Inductive/Transductive learning on PPI: no L2 regularization, no dropout, 3-layer GAT with residual connections added for the last two layers; a learning rate of 0.005 for *concat* attention and a learning rate of 0.0001 for other attention variants; the number of attention

---

**Algorithm 1** Random Walk Sampling

---

**Require:** $\mathcal{G} = (\mathcal{V}, \mathcal{E})$ the original graph, $g\_size = 100$ the target subgraph size
 1: $step = 0$
 2: $start \sim Unif(\mathcal{V})$                          ▷ Uniformly choose a starting node.
 3: $\mathcal{V}_{sub} = \{start\}, \mathcal{E}_{sub} = \{(start, start)\}$
 4: $src = start$
 5: **while** $|\mathcal{V}_{sub}| < g\_size$ and $step < 100 * g\_size$ **do**
 6:      $step = step + 1$
 7:      $back \sim Bernoulli(0.15),$           ▷ Return to the starting point with probability 0.15.
 8:      **if** back **then**
 9:          $src = start$
10:      **else**
11:          $dst \sim Unif(\{j|(src, j) \in \mathcal{E}\})$
12:          $\mathcal{V}_{sub} = \mathcal{V}_{sub} \bigcup \{dst\}$
13:          $\mathcal{E}_{sub} = \mathcal{E}_{sub} \bigcup \{(dst, dst), (src, dst), (dst, src)\}$
14:          $src = dst$
15: Return $(\mathcal{V}_{sub}, \mathcal{E}_{sub})$

---

     heads in the three layers is separately $8, 8, 6$ for *general* attention and $4, 4, 6$ for other attention variants; 128 hidden units for each head in the first two layers for *general* attention and 256 hidden units for each head in other attention variants[1]; we use a batch size 1 for *general* attention and a batch size 2 for other attention variants; binary cross entropy loss

- Inductive learning on Cora, Citeseer, Pubmed: batch size 24, 3-layer GAT with separately $4, 4, 6$ attention heads, residual connection is added for the last two layers, 8 hidden units per head for the first two layers, a learning rate of $0.005$, a dropout of $0.6$; L2 regularization with coefficient $0.001$ for pubmed and $0.0005$ for the rest two citation networks; cross entropy loss

- CEP: a batch size of $512$, 3-layer GAT where each layer has 4 heads and each head has 32 hidden units, a dropout of $0.0$, residual connection is added for the last two layers, a learning rate of $0.001$, no L2 regularization; smooth L1 loss

- HIV: a batch size of $64$, 2-layer GAT where each layer has 4 heads and each head has 32 hidden units, a dropout of $0.0$, residual connection is added for the last two layers, no L2 regularization, an initial learning rate of $0.0005$, a decay of learning rate by $0.99$ after each epoch; weighted focal loss with $-(wy(1-p)^{\gamma} \log p + (1-y)p^{\gamma} \log (1-p))$, where $\gamma = 2$ and $w = \#$negative samples$/\#$positive samples

An early stop is performed if the validation score hasn't been improved for 100 epochs.

### B.3 GRAPH-LEVEL PREDICTION

Based on node features updated with a GNN, we can also perform a graph-level prediction. First, a graph representation can be obtained with:

$$h_G = \sum_{v \in \mathcal{V}} \text{Sigmoid}\left(g\left(h_v^L\right)\right) \text{ReLU}(f\left(h_v^L\right))$$

where $L$ is the number of GNN layers, $g : \mathbb{R}^{n_L} \to \mathbb{R}$ and $f : \mathbb{R}^{n_L} \to \mathbb{R}^{n_G}$ are two linear layers with bias added. In all cases we consider $n_G = 128$. A graph-level prediction is then computed with a 3-layer MLP where all hidden sizes are equal to $n_G$ and a ReLU activation is applied after each of the first two linear layers.

---

[1]With the formulation of $\text{score}(h_i, h_j) = (h_i)^T B h_j$, the *general* attention variant requires a lot more parameters than the other two attention variants with a same number of hidden units, which can result in an out of memory error. As a work around, we use a larger number of heads for this variant with a smaller hidden size per head so that the final output size of layers does not change.

Table 8: GAT performance with three attention types

| Datasets | Reference | Concat | General | Dot product |
|---|---|---|---|---|
| Cora | $83.0 \pm 0.7\%$ | $83.0 \pm 0.7\%$ | $\mathbf{84.2 \pm 0.5}\%$ | $84.0 \pm 0.5\%$ |
| Citeseer | $\mathbf{72.5 \pm 0.7}\%$ | $\mathbf{72.5 \pm 0.7}\%$ | $71.5 \pm 0.9\%$ | $71.4 \pm 0.8\%$ |
| Pubmed | $\mathbf{79.0 \pm 0.3}\%$ | $\mathbf{79.0 \pm 0.3}\%$ | $78.6 \pm 0.0\%$ | $78.2 \pm 0.7\%$ |
| PPI | $0.973 \pm 0.00$ | $0.973 \pm 0.00$ | $\mathbf{0.982 \pm 0.00}$ | $0.975 \pm 0.00$ |
| PPI trans $5\%$ | | $0.476 \pm 0.03$ | $\mathbf{0.565 \pm 0.01}$ | $0.524 \pm 0.01$ |
| PPI trans $79\%$ | | $\mathbf{0.950 \pm 0.00}$ | $0.936 \pm 0.01$ | $0.947 \pm 0.00$ |
| Cora inductive | | $87.6 \pm 1.7\%$ | $88.1 \pm 1.7\%$ | $\mathbf{88.4 \pm 1.3}\%$ |
| Citeseer inductive | | $84.2 \pm 0.9\%$ | $84.1 \pm 1.5\%$ | $\mathbf{84.8 \pm 1.3}\%$ |
| Pubmed inductive | | $85.3 \pm 1.1\%$ | $\mathbf{86.5 \pm 1.2}\%$ | $86.0 \pm 0.8\%$ |
| CEP | $0.66 \pm 0.12$ (Ryu et al., 2018)[2] | $0.43 \pm 0.02$ | $\mathbf{0.39 \pm 0.02}$ | $0.44 \pm 0.02$ |
| HIV | $\mathbf{0.776}$ (Li et al., 2017) | $0.746 \pm 0.02$ | $0.760 \pm 0.01$ | $0.758 \pm 0.02$ |

### B.4 TEST PERFORMANCE ACROSS ATTENTION VARIANTS

We evaluate test performance using different metrics for different datasets – accuracy for *Cora*, *Citeseer*, *Pubmed*, micro-averaged F1 score for *PPI*, mean absolute error for *CEP* and roc auc score for *HIV*. See table 8 for a summary of the prediction performance, where different attention variants mostly have similar performance. The reference numbers are from Veličković et al. (2018) unless stated otherwise. For *Cora*, *Citeseer*, *Pubmed* and *PPI*, we include the original results of GATs for reference. For the rest datasets, we include the best performance of previous work for reference whenever applicable, but some models do not involve attention mechanism.

### B.5 GRAPH CLASSIFICATION

**Dataset construction** 1) As PPI has multiple graphs, we sample the same number of subgraphs from each original graph. 2) For CEP dataset, we first sort the whole dataset based on the photovoltaic efficiency and split it into 96 buckets. We sample 5 graphs from each bucket where 3 graphs are used for training, one graph is used for validation and one graph is used for test. 3) As the HIV dataset is intrinsically imbalanced, we construct a balanced subset by sampling a same number of positive samples and negative samples. Also since we are considering a special dataset split, the training, validation and test subset is separately constructed from the training, validation and test set. 4) For Cora, Citeseer, Pubmed and PPI, we construct a whole dataset of subgraphs first and then perform a random split to get the training, validation and test set with a splitting ratio of 60%:20%:20%.

**Graph feature extraction** For graph feature extraction, we train a 2-layer GAT with 8 heads for each layer and 64 hidden units for each head. We use a batch size of 16 and perform an early stop if the validation score no longer improves for 10 epochs. We use the same loss functions for each dataset as explained in B.2. We also perform a hyperparameter search for learning rate, dropout, L2 regularization coefficient $\lambda$ and whether to perform a residual connection. The selected hyperparameters are as follows:

- Cora: a dropout of $0.1$, residual connection is added for the second layer, no L2 regularization, a learning rate of $0.01$ for *concat* attention and a learning rate of $0.005$ for other attention variants

- Citeseer: a dropout of $0.1$, residual connection is added for the second layer, a learning rate of $0.01$, no L2 regularization

- Pubmed: a dropout of $0.1$, residual connection is added for the second layer, no L2 regularization, a learning rate of $0.005$ for *general* attention and a learning rate of $0.01$ for other attention variants

- PPI: no dropout, residual connection is added for the second layer, no L2 regularization, a learning rate of $0.005$ for *concat* attention and a learning rate of $0.01$ for other attention variants

---

[2]The original work only has a bar plot and we contacted the authors for the numbers.

- CEP: no dropout, a learning rate of $0.005$; a residual connection is added for the second layer only with *general* attention; L2 regularization with coefficient $0.001$ is used except for *concat* attention

- HIV: a residual connection is added for the second layer, no L2 regularization is used; a dropout of $0.6$ is used only for *concat* attention; a learning rate of $0.005$ is used for *dot product* attention and a learning rate of $0.01$ is used for the rest attention variants

**t-SNE visualization of attention metrics**   In the text we included the t-SNE visualization of attention based features for *concat* attention only. Here we include the results for all three variants for comparison in figure 7, 8 and 9. Across all attention variants, we observe a similar pattern that the attention metrics for citation subgraphs get blurred while those for the rest datasets are better separated and clustered.

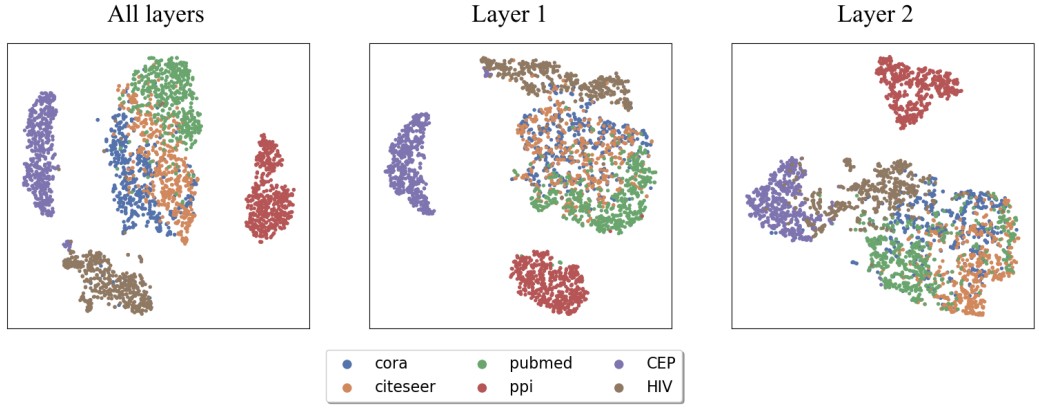

Figure 7: t-SNE visualization of *concat* attention based features. From left to right, the features are separately from all layers, the first layer and the second layer.

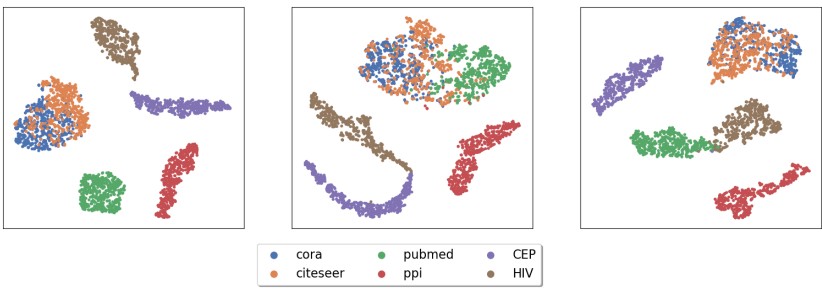

Figure 8: t-SNE visualization of *dot product* attention based features. From left to right, the features are separately from all layers, the first layer and the second layer.

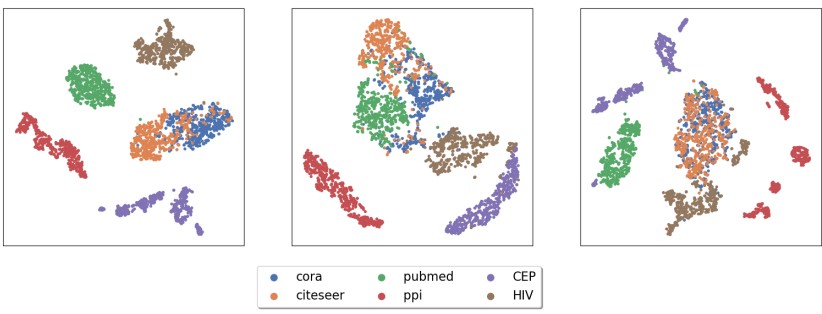

Figure 9: t-SNE visualization of *general* attention based features. From left to right, the features are separately from all layers, the first layer and the second layer.

