# OpenReview forum: "Characterize and Transfer Attention in Graph Neural Networks"
_ICLR.cc/2020/Conference — Reject_

### Official Review · AnonReviewer1 · 2019-10-22
**Official Blind Review #1**

**Rating:** 6

**Review:**

This paper presents an empirical study of the attention mechanism in the graph attention networks (GAT).  The study reveals that the attention patterns largely depend on the dataset, on some datasets they are sharp, but on others the attention patterns are almost uniform and not so different from the uniform aggregation weights in GNNs that does not have attention.  The authors further tried to utilize these findings and attempted to do attention-based graph sparsification, and showed that they can get a similar level of performance with only a fraction of the edges in the original graph if they do the sparsification based on the attention weights.

Given the popularity of the GAT model in the graph neural networks literature and the effectiveness of the attention mechanism in many deep learning architectures, the empirical study presented in this paper focused on attention is valuable.  The experiments are clearly motivated and executed, which I appreciate.

As this is an empirical paper, one (maybe) problem with it is that the findings presented aren’t that surprising in hindsight - of course the attention patterns should be data dependent, and doing attention-based graph sparsification seems like an obvious thing that should work.  The results on dataset-dependent attention patterns may have told us more about the datasets rather than the GAT model.

There are a few presentation issues that need clarification:
- sec 5.1: it is not clear from the text what \alpha^{static} is.  As mentioned earlier there are multiple possible static attention weights (GCN vs GraphSAGE).
- sec 5.1: I found it strange to normalize the discrepancy score by 2|V|.  The number of terms in the sum should be 2|E| where |E| is the number of edges as each edge is counted twice.  Normalizing by 2|V| does not guarantee the score is in [0, 1] as claimed in the paper.
- sec 5.1: “Besides, these attention do not get concentrated on self loops based on relatively stable values.” -- I don’t see why having stable attention values can show there’s no concentration on self-loops.
- Figure 5 left: looks like the curves can’t reach the right end, this means 1 <= k <= 8 is probably not a good range.
- Table 5 is a bit confusing.  From the text my understanding is that a GAT is trained first, and then do sparsification, and then train another GraphSAGE on the sparsified graph to do prediction.  It’s not immediately clear why the GAT performance in Table 5 is so bad, while the GraphSAGE performance is just way better.  After reading this a few times I realized a much smaller GAT is used (with much smaller hidden size) while the GraphSAGE model is always using a large hidden size.  I think this part needs some improvement.

Overall I liked the empirical study and think the community can benefit from this paper.

**Experience Assessment:**

I have published in this field for several years.

**Review Assessment: Checking Correctness Of Derivations And Theory:**

N/A

**Review Assessment: Checking Correctness Of Experiments:**

I carefully checked the experiments.

**Review Assessment: Thoroughness In Paper Reading:**

I read the paper thoroughly.

---

> ### Author Response · Authors · 2019-11-12
> **Re: Official Blind Review #1**
>
> We thank the reviewer for the positive feedback and constructive suggestions on presentation improvement.
>
> 1. You are right that the proposed metric can measure the discrepancy between any static attention (GCN or GraphSAGE) and the learned attention. We've added a note to make it more clear.
>
> 2. Given a pair of probability distributions, the L1 distance between probabilities is in the range of [0, 2]. For example, you may consider two Bernoulli distributions where they place probability 1 on two different values. Dividing this distance by 2 gives us a range of [0, 1]. Now with |V| nodes, we have |V| pairs of distributions in total and dividing the total sum by |V| gives back the range [0, 1].
>
> 3. You are right that the expression is a bit ambiguous. The motivation is that we want to see if attention gets more concentrated on self loops while getting sharper, suggesting that GNNs are degenerating to MLPs. We've modified the expression and it now says "Besides, the attention does not get increasingly more concentrated on self loops while getting sharper over layers."
>
> 4. For the left subfig of Figure 5, we guess you are probably referring to blue curves while results of top-k sparsification based on learned attention are plotted with orange curves. The reason for the big gap between blue curves and the original result is that the PPI graphs have some hub nodes with extremely large degree. In such cases, uniform neighbor sampling is not very effective after a certain degree.
>
> 5. We've tried to make the description more clear and made an update.

---

> > ### Comment · AnonReviewer1 · 2019-11-12
> > **Re: Re: Official Blind Review #1**
> >
> > Thanks for taking the time to address my questions.
> >
> > Regarding your response number 2, either I misinterpreted your terminology, or I think your explanation about normalizing by 2|V| is still incorrect, in a few ways:
> >
> > (1) The L1 distance between two probability values lies in the range of [0, 1], not [0, 2].  If both a and b are in the range of [0, 1], then their difference |a-b| cannot be greater than 1.
> >
> > (2) You have a double sum, sum_{i in V} sum_{j in N(i)} |alpha_{i,j}^learned - alpha_{i,j}^static|, which is firstly a sum over nodes, and then for each node a sum over its neighbors, effectively the number of terms in this double sum is the number of edges in the graph, multiplied by 2.  So if you normalize by 2|V| this quantity is not guaranteed to be in [0, 1], but if your normalize by 2|E| it would.

---

> > > ### Author Response · Authors · 2019-11-12
> > > **Re: Re: Re: Official Blind Review #1**
> > >
> > > See if the elaboration below makes sense.
> > >
> > > Given a node $i$, the attention value of $i$ over its one-hop neighbors $\{\alpha_{i,j}\}_{j\in\mathcal{N}(i)}$ forms a probabilitty distribution over $\mathcal{N}(i)$.
> > >
> > > With static attention and learned attention, we have two distributions $\{\alpha_{i,j}^{learned}\}_{j\in\mathcal{N}(i)}$ and $\{\alpha_{i,j}^{static}\}_{j\in\mathcal{N}(i)}$. The L1 distance between them is then $\sum_{j\in\mathcal{N}(i)}|\alpha_{i,j}^{learned}-\alpha_{i,j}^{static}|$, which is in the range of $[0, 2]$. For example, 2 is achieved if the two types of attention place $1$ on two different neighbors ($|1-0| + \cdots + |0-1| + \cdots$). By dividing it by $2$, we get a range of $[0, 1]$.
> > >
> > > Finally, we go through all nodes, and take an average, which gives us $\frac{1}{|\mathcal{V}|}\sum_{i\in \mathcal{V}}$.
> > >
> > > The approach you suggested is also very interesting and seems equivalent. I think your approach places an emphasis over full graphs while my motivations here root in one-hop neighborhoods.

---

> > > > ### Comment · AnonReviewer1 · 2019-11-13
> > > > **Re: Re: Re: Re: Official Blind Review #1**
> > > >
> > > > Thanks for the clarification.  This is now clear.  For a reader that is more used to the $\sum_{i\in\mathcal{V}}\sum_{j\in\mathcal{N}(i)}$ notation being a sum over all edges this is a bit unintuitive.  It would be good to add some explanation of this in the paper.

---

> > > > > ### Author Response · Authors · 2019-11-13
> > > > > **Re: Re: Re: Re: Re: Official Blind Review #1**
> > > > >
> > > > > Thank you for the suggestion and I've made an update.

---

### Official Review · AnonReviewer2 · 2019-10-24
**Official Blind Review #2**

**Rating:** 3

**Review:**

This paper analyzes attention in graph neural networks. It makes two major claims:
(1) Datasets have a strong influence on the effects of attention. The attention in citation networks are more uniform, but they behave differently in protein or molecule graphs.
(2) With attention-based graph sparsification, it is possible to remove a large portion of edges while maintaining the performance.

I have some concerns about this paper: (1) the analysis lacks theoretical insights and does not seem to be very useful in practice; (2) the proposed method for graph sparsification lacks novelty and the experiments are not thorough to validate its usefulness; (3) the writing of this paper is messy, missing many details.

In the analysis part (section 5), the choices of probing metrics seem arbitrary and lack theoretical insights. The authors used the L1 norm, but it seems not appropriate for the tasks here, e.g. KL divergence is preferred to measure the distributional discrepancy, entropy for concentration etc. Many important details are missing or not clear, for example, in Table 2, which head/layer is used for computing the attention, and what does “GCN vs learned” mean? The maximum pairwise difference is not clearly defined. The meta graph classification (section 5.2) only considers a synthetic dataset. Overall, I feel the analysis didn’t present too many interesting observations, and I cannot see too much potential value in applications (even for the graph sparsification task in this paper, its correlation with the analysis is quite weak).

In section 6, it explores whether it is possible to remove part of the edges from the graph while maintaining the classification performance. It is an interesting task, but the method proposed in this paper is not realistic and lacks novelty. In 6.1 and 6.2, it needs to train a GAT first to get the attention scores, then remove edges according to attention scores and train another GAT. In this way, it doesn’t reduce the computational requirement, as it still trains a full model to get the attention. Only in 6.3 it presents a realistic setting, where the attention scores are derived from a small GAT, and train another GNN on the sparsified graph. But the paper didn’t explain why it is possible to get reliable attention scores with a small GAT, and the experiment is only on one dataset. Does it apply to other datasets (citation network, molecules) and settings (transductive, inductive)? So far the experiments are not enough to be considered as a valid contribution.

**Experience Assessment:**

I have published one or two papers in this area.

**Review Assessment: Checking Correctness Of Derivations And Theory:**

N/A

**Review Assessment: Checking Correctness Of Experiments:**

I assessed the sensibility of the experiments.

**Review Assessment: Thoroughness In Paper Reading:**

I read the paper at least twice and used my best judgement in assessing the paper.

---

> ### Author Response · Authors · 2019-11-12
> **Re: Official Blind Review #2**
>
> We thank the reviewer for pointing out multiple directions for future work. Below we make some clarifications:
>
> 1. We choose L1 distance (also known as total variation when considering probability distributions) over KL divergence because it has a bounded range of $[0, 1]$ while KL divergence can be unbounded. To have an upper bound can make it easier for users to understand how numbers get associated with discrepancy. We've also considered entropy, but node degree can have a strong effect on the entropy of attention distribution, which is not easy to decouple.
>
> 2. "in Table 2, which head/layer is used for computing the attention" -- we've updated the title of the table to make it more clear.
>
> 3. "GCN vs learned" means the discrepancy between the static GCN attention (defined in 3.1) and the learned attention.
>
> 4. "The maximum pairwise difference is not clearly defined." -- We've added an equation to make the definition more clear.
>
> 5. Graph sparsification is not very meaningful when the attention is almost uniform. Molecules are already very sparse. We agree it will be better to have more applicable datasets for verification.

---

### Official Review · AnonReviewer3 · 2019-10-24
**Official Blind Review #3**

**Rating:** 1

**Review:**

This paper carries out several kinds of analysis on the GAT networks of Velickovic (2018), which augment GNN updates with multihead self attention. Three standard attention types are compared, on several different datasets, and differences between uniform attention and learned attention are reported. An experiment is carried out where low-attention edges are pruned.

While understanding the value of attention is important, this paper leaves many questions open. First, since the graphs studied in this paper are, if not generally sparse to begin with at least they only include connections that are meaningful, the sparsification experiment is a bit hard to understand. One particular extension would improve things: adding random edges (can the model learn to prune them out?), but learning sparse attention (see e.g., Maruf et al., 2019) rather than thresholding seems to be a reasonable point of comparison.

Overall this paper would be more valuable if a clear and concise recommendation could be given regarding how to use or understand attention; but the lack of a consistent pattern of results makes any obvious narrative hard to support. I would encourage the authors to continue this line of work so that it can be used to provided guidance to those who would like to make more effective use of GNNs.

**Experience Assessment:**

I have published one or two papers in this area.

**Review Assessment: Checking Correctness Of Derivations And Theory:**

N/A

**Review Assessment: Checking Correctness Of Experiments:**

I assessed the sensibility of the experiments.

**Review Assessment: Thoroughness In Paper Reading:**

I read the paper at least twice and used my best judgement in assessing the paper.

---

> ### Author Response · Authors · 2019-11-12
> **Re: Official Blind Review #3**
>
> We thank the reviewer for his encouragement and suggestions on future work.

---

### Decision · Program_Chairs · 2019-12-19

**Decision:**

Reject

**Comment:**

This paper suggests that datasets have a strong influence on the effects of attention in graph neural networks and explores the possibility of transferring attention for graph sparsification, suggesting that attention-based sparsification retains enough information to obtain good performance while reducing computational and storage costs.

Unfortunately I cannot recommend acceptance for this paper in its present form. Some concerns raised by the reviewers are: the analysis lacks theoretical insights and does not seem to be very useful in practice; the proposed method for graph sparsification lacks novelty; the experiments are not thorough to validate its usefulness. I encourage the authors to address these concerns in an eventual resubmission.